# Diagnostic Performance of ^18^F-Choline Positron Emission Tomography/Contrast-Enhanced Computed Tomography in Adenoma Detection in Primary Hyperparathyroidism after Inconclusive Imaging: A Retrospective Study of 215 Patients

**DOI:** 10.3390/cancers14082029

**Published:** 2022-04-17

**Authors:** Johan Benjamin, Laure Maillard, Isabelle Morelec, Philippe Got, Françoise Borson-Chazot, Jean-Christophe Lifante

**Affiliations:** 1Service de Chirurgie Endocrinienne, Centre Hospitalier Lyon Sud, Hospices Civils de Lyon, 69495 Pierre-Bénite, France; laure.maillard@chu-lyon.fr; 2Service de Médecine Nucléaire, Centre Hospitalier Lyon Sud, Hospices Civils de Lyon, 69495 Pierre-Bénite, France; isabelle.morelec@chu-lyon.fr (I.M.); philippe.got@chu-lyon.fr (P.G.); 3Fédération d’Endocrinologie, Hôpital Louis Pradel, 69500 Bron, France; francoise.borson-chazot@chu-lyon.fr

**Keywords:** ^18^F-choline PET/CT, primary hyperparathyroidism, parathyroid adenoma, minimally invasive surgery, diagnostic performance

## Abstract

**Simple Summary:**

Primary hyperparathyroidism is a common pathology. Its curative treatment is based on surgery with precise localisation of the affected parathyroid glands. Our retrospective study aimed to assess the diagnostic performance of a novel imaging method called ^18^F-choline positron emission tomography/contrast-enhanced computed tomography (PET/ceCT) in adenoma detection after inconclusive imaging. ^18^F-choline PET/ceCT presented excellent diagnostic performance as a second-line imaging method. These results confirm its utility, and it could replace Tc99m-sestamibi single photon emission CT/CT as the first-line imaging method in patients with primary hyperparathyroidism.

**Abstract:**

This large, retrospective, single-centre study evaluated the diagnostic performance of ^18^F-choline positron emission tomography/contrast-enhanced computed tomography (PET/ceCT) in preoperative parathyroid adenoma detection in primary hyperparathyroidism cases after negative/inconclusive ultrasound or other imaging findings. We included patients who underwent surgery and ^18^F-choline PET/ceCT for inconclusive imaging results between 2015 and 2020. We compared the ^18^F-choline PET/ceCT results with surgical and histopathological findings and identified the variables influencing the correlation between ^18^F-choline PET/ceCT and surgical findings. Of 215 enrolled patients, 269 glands (mean lesion size, 10.9 ± 8.0 mm) were analysed. There were 165 unilocular and 50 multilocular lesions; the mean preoperative calcium level was 2.18 ± 0.19 mmol/L. Among 860 estimated lesions, 219 were classified as true positive, 21 as false positive, and 28 as false negative. The per-lesion sensitivity was 88.66%; specificity, 96.57%; positive predictive value, 91.40%; and negative predictive value, 95.39%. The detection and cure rates were 82.0% and 95.0%, respectively. On univariate and multivariate analyses, the maximum standardised uptake value (SUVmax), lesion size, and unilocularity correlated with the pathologic findings of hyperfunctioning glands. ^18^F-choline PET/ceCT presents favourable diagnostic performance as a second-line imaging method, with SUVmax, lesion size, and unilocularity predicting a high correlation between the ^18^F-choline PET/ceCT and surgical findings.

## 1. Introduction

Primary hyperparathyroidism is a frequent endocrine pathology. Its curative treatment is chiefly based on surgery, which is guided by preoperative imaging that provides information on the location and pathological aspects of the parathyroid glands. The two most commonly used examination techniques are cervical sonography and Tc99m-sestamibi (MIBI) tomoscintigraphy. Both these examinations have good diagnostic performances [1] and are the standard examinations in the first-line assessment. In case of good concordance between these two imaging methods, minimally invasive surgery is proposed, which presents a reduced complication rate and operative time with a satisfactory cure rate [2,3,4,5,6].

However, these examination approaches have limitations. Cervical sonography is an operator-dependent examination requiring extensive experience in this type of pathology [7,8]. Moreover, some ectopic locations, such as paraoesophageal or intramediastinal areas, cannot be explored, leading to false negative results [9]. The limitations of MIBI scintigraphy include reduced sensitivity in cases with multiglandular or small lesions and low levels of parathyroid hormone (PTH) [10,11].

Thus, in cases of proven hyperparathyroidism with discordant or inconclusive imaging results, surgical exploration is still required, which exposes the patient to unnecessary or ineffective surgery and its severe complications.

Since 2013, a new parathyroid imaging method has been increasingly employed for the characterisation and tracking of parathyroid glands called ^18^F-choline positron emission tomography/contrast-enhanced computed tomography (PET/ceCT). This approach has conventionally been used for the evaluation of prostate cancer or hepatocellular carcinoma [12,13], and its usefulness in the detection of parathyroid adenoma was discovered accidentally [14,15]. Since then, its use has been on the rise, particularly in cases with inconclusive or discordant imaging findings. Various studies have demonstrated the good performance of this method with detection rates ranging from 88% to 100% [16,17,18,19,20,21] and sensitivity (Se) of approximately 89% [22]. Nevertheless, these numbers are based on small retrospective studies, limiting their extrapolation to real-life use.

This study aimed to determine the diagnostic performance of ^18^F-choline PET/ceCT in a large population of patients with primary hyperparathyroidism with negative or inconclusive imaging results. Additionally, we aimed to identify the factors influencing the correlation between ^18^F-choline PET/ceCT and surgical findings.

## 2. Materials and Methods

### 2.1. Population

This retrospective study, conducted at an academic department of endocrine surgery, aimed at assessing the diagnostic performance of ^18^F-choline PET/ceCT for the localisation of parathyroid adenoma or hyperplasia in patients with primary hyperparathyroidism.

We included all patients aged >18 years who underwent ^18^F-choline PET/ceCT for adenoma localisation and parathyroidectomy between January 2015 and May 2020. During this period, inconclusive or negative imaging results were the only indication for ^18^F-choline PET/ceCT for parathyroid adenoma research. All excised glands were analysed to assess their pathological state.

We excluded all patients aged <18 years and those who underwent ^18^F-choline PET/ceCT after the surgery.

In accordance with the French ethical directives, the need for written informed consent from the participants was waived (MR 004).

### 2.2. Measurements

#### 2.2.1. General Measurements

For every patient we retrieved data for the following general parameters:Demographic data: sex, age, body mass index (BMI)Medical background: hypertension, diabetes, osteoporosis, nephrolithiasis, multiple endocrine neoplasia (MEN), *HRPT2* mutation, and persistent or recurrent hyperparathyroidismSurgical background: thyroid surgery or parathyroid surgery

#### 2.2.2. Preoperative Measurements

For every patient and lesion, we retrieved the following data:Biological findings: calcium level (mmol/L) and PTH level (expressed as the PTH/normal PTH ratio due to differences in normal ranges among laboratories)Preoperative ^18^F-choline PET/ceCT data: lesion location, number of lesions, lesion size, and maximum standardised uptake value (SUVmax)

#### 2.2.3. Perioperative Findings

For every patient and lesion, we collated information on the operator, surgical approach (standard bilateral parathyroid exploration or minimally invasive), associated thyroidectomy, and the number of adenomas and their locations.

#### 2.2.4. Histological Findings

Every removed gland was subjected to a definitive anatomopathological examination to confirm the pathological state. A gland was defined as pathological if a hyperplasic or adenoma-type structure was found. In cases of associated thyroid removal, histological examination was performed.

#### 2.2.5. Follow-Up

All patients were discharged after 24 h if there were no complications and the calcium levels had normalised compared to the calcium level and PTH ratio on day 1. Subsequently, we assessed the calcium level and PTH ratio at week 6, and the patient was considered cured if the calcium level had normalised during this postoperative visit.

### 2.3. Protocols

#### 2.3.1. ^18^F-Choline PET/ceCT Protocol

For imaging acquisition, we used a Discovery Dual PET/CT camera (General Electric) with low-dose CT (120 kVp, 30–50 mAs). Each patient received 3 MBq/kg of ^18^F-choline, and CT imaging of the neck and upper mediastinum was performed 20 min after the injection, followed immediately by PET imaging of the same area (2 min per bed position). A second acquisition was performed immediately afterward, with the field of view covering the neck (4 min per bed position). During this phase, CT was performed with intravenous administration of a contrast medium. PET datasets were reconstructed using a time-of-flight reconstruction with ordered subset expectation maximisation using two iterations and 24 subsets. All scans were independently interpreted on-site by two experienced nuclear medicine physicians who were blinded to the ultrasonography and scintigraphy results. These experts visually evaluated the number of foci indicative of hyperfunctioning parathyroid glands and their respective locations. They also determined the SUVmax for any abnormal foci.

#### 2.3.2. Surgical Protocol

Two expert surgeons performed the surgical procedures. The surgical approach was either the standard bilateral parathyroid exploration or a minimally invasive procedure. The choice was left at the surgeon’s discretion depending on different factors (recurrence, osteoporosis, and concordance with cervical sonography findings).

A 6-cm Kocher incision was used for the standard bilateral parathyroid exploration. After superficial dissection of the platysma muscle, an incision between the two infrahyoid muscle groups permitted the exploration of the different parathyroid gland locations. All glands underwent identification attempts, and only the pathological glands were removed at the end of the dissection.

For minimally invasive procedures, a 3-cm incision was made. Minimal invasion does not necessarily mean a small scar but rather explores only one parathyroid gland according to the preoperative imaging results. In this case, only the pathologic gland was explored, guided by preoperative imaging.

We conducted a systematic extemporaneous examination to confirm the parathyroid nature of the excised tissue. We did not use in situ fluorescence imaging or assess the intra-operative PTH levels.

For the ectopic glands, different surgical approaches were used as required, such as thoracoscopy or manubriotomy, especially in cases with profound intramediastinal glands.

### 2.4. Data Interpretation and Statistics

#### 2.4.1. Diagnostic Performances

The gold standard for diagnosis was the presence of a gland with pathological findings (adenoma or hyperplasia) and its removal resulting in a cure of the condition at week 6. For every gland, we defined the following:True positive (TP): a removed gland leading to cure with pathological findings on histological examination.False positive (FP): a removed gland not leading to cure but with normal findings on histological examination.False negative (FN): a removed gland not considered pathological on ^18^F-choline PET/ceCT but its removal leads to cure with pathological findings on histological examination.True negative (TN): glands not considered pathological on ^18^F-choline PET/ceCT or during surgical exploration.

Thus, we were able to calculate the Se, specificity (Sp), positive predictive value (PPV), and negative predictive value (NPV) in the per-lesion and per-patient analyses.

#### 2.4.2. Correlation between ^18^F-Choline PET/ceCT and Surgical Findings

We then studied the correlation between the ^18^F-choline PET/ceCT and surgical findings for every gland. All the removed glands with pathological findings on histological examination showing identical locations during ^18^F-choline PET/ceCT and surgical examinations were considered to have a high correlation.

On the other hand, the uncorrelated entities were as follows:Removed glands with different locations on ^18^F-choline PET/ceCT and during a surgical examinationRemoved glands not described as pathological on ^18^F-choline PET/ceCT but showing pathological findings on histological examinationRemoved glands described as pathological on ^18^F-choline PET/ceCT but without pathological findings during surgical and/or histological examination

#### 2.4.3. Statistics

Descriptive analyses of the study population were performed. Qualitative values are presented as numbers and percentages. Quantitative values are presented as means ± standard deviations and ranges.

We then performed an explicative analysis by conducting univariate and multivariate analyses of the factors influencing the correlations. The χ^2^ and Fisher’s exact tests were used to analyse categorical variables depending on the size of the population in the univariate analysis. Multivariate analysis was performed using three different mixed generalised models for SUVmax and lesion size in per-lesion analysis and logistic regression for the number of lesions in per-patient analysis adjusted for age, BMI, osteoporosis, preoperative calcium level, and PTH ratio. *p*-values < 0.05 were considered statistically significant.

All data were analysed using R Software (R Foundation for Statistical Computing, Vienna, Austria) and pvalue.io (Medistica; pvalue.io is a graphic user interface to the R statistical analysis software for scientific medical publications 2019, https://www.pvalue.io/fr (accessed on 16 June 2021)).

## 3. Results

### 3.1. Population Characteristics

The population characteristics are summarised in Table 1. In total, 215 patients (52 men and 163 women) from March 2015 to February 2020 were included in the analysis. Their mean age was 62.0 ± 14.2 (18–87) years. The presence of associated pathologies was common; 81 (38%) patients presented with hypertension, 84 (38%) with osteoporosis, 27 (13%) with fractures, and 63 (29%) with nephrolithiasis. Among the total patients, 14 (6.6%) had undergone previous thyroid surgery, 20 (9.3%) had recurrent hyperparathyroidism, 10 (4.7%) had persistent hyperparathyroidism, seven (3.3%) had undergone previous minimally invasive parathyroidectomy, and 23 (11%) had undergone previous standard bilateral cervical exploration. The mean preoperative calcium level was 2.77 ± 0.19 (2.28–3.64) mmol/L, whereas the mean preoperative PTH ratio was 2.18 ± 1.0 (0.61–19.8). Four (1.9%) patients presented with MEN type 1, and one (0.47%) had *HRPT2* mutation.

### 3.2. Surgical Results

During the study period, 260 glands were removed. Examination of a single gland was performed in 165 (76.7%) patients, two glands in 46 (21.4%) patients, and three glands in four (1.9%) patients. Concerning their localisation, 19 glands were found in major ectopic positions, such as intrathymic, retrosternal, or within the vagal nerve or carotid sheath. Regarding the surgical approach, 163 (76%) patients underwent standard bilateral parathyroid exploration, whereas 52 (24%) underwent minimally invasive surgery. Four sternotomies and one thoracoscopy were conducted for the major ectopic adenomas. With respect to complications, six (2.9%) patients experienced recurrent laryngeal nerve palsy and seven (3.4%) had hypocalcaemia requiring the use of alfacalcidol. The average postoperative calcium levels at day 1 and week 6 were 2.38 ± 0.196 (1.93–3.12) mmol/L and 2.37 ± 0.14 (1.19–2.91) mmol/L, respectively, while the postoperative PTH ratios were 0.35 ± 0.26 (0.10–1.94) and 0.92 ± 0.47 (0.28–4.63), respectively. The cure rate at 6 weeks was 95%.

The histological findings revealed 188 (72.3%) adenomas and 46 (17.7%) hyperplasias. Furthermore, the following unexpected histological results were found: one parathyroid carcinoma, one thymoma, and three thyroidal nodules.

### 3.3. ^18^F-Choline PET/ceCT Results and Performances

^18^F-choline PET/ceCT detected 240 glands as having pathological findings. The mean SUVmax was 3.76 ± 2.34 (0–18.5) and the mean lesion size was 10.70 ± 7.99 (0–53) mm. The ^18^F-choline PET/ceCT revealed two pathologic glands in 31 patients, three pathologic glands in three patients, and no pathologic glands in 10 patients. In the per-lesion analysis, 219 glands were classified as TP (Figure 1), 21 as FP (Figure 2), 28 as FN (Figure 3), and 592 as TN. In the per-patient analysis, 164 patients were classified as TP, 21 as FP, and 25 as FN. Furthermore, the per-lesion and per-patient analyses of ^18^F-choline PET/ceCT showed Se of 88.6% and 86.8% and PPV of 91.4% and 88.6%, respectively. The Sp and NPV in the per-lesion analysis were 96.6% and 95.4%, respectively. The ^18^F-choline PET/ceCT results are summarised in Table 2.

### 3.4. Correlation between ^18^F-Choline PET/ceCT and Surgical Findings

Table 3 summarises the results of 269 lesions. In total, 240 (81.8%) lesions were classified as ‘well correlated’ and 49 (18.2%) as ‘uncorrelated’. The general characteristics of the populations were not statistically different between the two groups. Similarly, the biological characteristics were not statistically different between the well-correlated and uncorrelated groups (preoperative calcium levels: 2.76 ± 0.214 vs. 2.77 ± 0.176 mmol/L, *p* = 0.67; PTH ratios: 2.27 ± 2.18 vs. 1.95 ± 0.947, *p* = 0.12). However, the surgical approach used differed between the two groups; minimally invasive surgery was not used in the uncorrelated group.

The ^18^F-choline PET/ceCT data were statistically different between the two groups. Lesions were significantly larger in the well-correlated group than in the uncorrelated group (12.2 ± 7.61 vs. 3.90 ± 6.02 mm, *p* < 0.001), and the SUVmax was significantly higher in the former than in the latter (4.25 ± 2.25 vs. 1.55 ± 2.00, *p* < 0.001). Moreover, the number of lesions detected by ^18^F-choline PET/ceCT was different between the two groups. In the well-correlated group, 162 (74%) lesions were unilocular compared to only 22 (45%) in the uncorrelated group (*p* < 0.001). Figure 4, Figure 5 and Figure 6 summarize the lesion size, SUVmax, and number of glands in the two groups, respectively.

The results of the multivariate analysis are summarised in Table 4. The odds ratios (95% confidence intervals) for SUVmax, lesion size, and number of lesions were 0.379 (0.28, 0.51), 0.703 (0.631, 0.785), and 0.151 (0.038, 0.49), respectively. The marginal effect of SUVmax was 9.8%, meaning that for every additional SUVmax point, the correlation was augmented by 9.8%. The marginal effect of lesion size was 3.3%, meaning that for every additional millimetre, the correlation was augmented by 3.3%. The marginal effect of the number of lesions was −18.8%, meaning that for every additional lesion detected, the correlation decreased by 18.8%.

## 4. Discussion

To the best of our knowledge, this is the largest study on the diagnostic performance of ^18^F-choline PET/ceCT in patients with inconclusive imaging findings for hyperparathyroidism [23,24]. The Se of this method was 88.6% in our study, which is consistent with previous results, especially with the prospective study by Quack et al. showing Se of 91.3% and PPV of 87.5% [22]. Few studies have evaluated the Sp, and it was 96.6% in our study after extrapolating the number of TNs. The excellent performance of ^18^F-choline PET/ceCT suggests its systematic use as the second option for adenoma localisation when the ultrasound and MIBI scintigraphy findings differ or when these methods cannot localize an adenoma.

We showed that the correlation between the ^18^F-choline PET/ceCT and surgical findings depends on the SUVmax, lesion size, and the number of pathological glands detected. These results suggest an extension of the minimally invasive approaches for single large glands with high SUV after inconclusive initial imaging, leading to reduced operative time and surgical risks.

Compared to MIBI, ^18^F-choline PET/ceCT seems better at adenoma detection. Lezaic et al. compared ^18^F-choline PET/ceCT with MIBI scintigraphy, and the former demonstrated a much higher Se (92% vs. 49%) [25]. Another advantage of ^18^F-choline PET/ceCT is the reduction in the examination duration from 3 h to 30 min. The short examination time of this method not only increases the comfort of the patients but also reduces the current price gap between ^18^F-choline PET/ceCT and MIBI scintigraphy by allowing more examinations within the same time. These results as well as the favourable performance recorded in our study support the replacement of MIBI scintigraphy with ^18^F-choline PET/CT. 

This imaging method combines functional imaging with the precise anatomical description for gland localisation. The use of ceCT in our protocol permitted the detection of at least two pathological glands without fixing ^18^F-choline. Thus, the results presented here combine the diagnostic performances of ^18^F-choline uptake and CT with injected contrast medium. These interesting results mimic those of another detection method, four-dimensional (4D) CT, which showed high performance, especially in patients with recurrent parathyroid disease [26,27]. Our protocol combining ceCT with PET had Se similar to that of 4D-CT, ranging from 89% to 93% [28,29]. The main difference between our ceCT protocol and the various 4D-CT protocols used is the lack of image acquisition during the arterial phase.

Cervical sonography tends to have highly variable Se and accuracy values (70–90% and 74–94%, respectively) [7,30,31], mostly due to its high dependence on the experience of the performing radiologist. The performance of ^18^F-choline PET/ceCT is comparable to that of cervical sonography when used as the first-line imaging approach. Moreover, CT scans provide a better guiding system for the surgeon performing the operation than cervical sonography, as the latter does not include a cervical section that can be analysed by the surgeon. This disadvantage may result in the use of ^18^F-choline PET/ceCT as a first-line imaging method for adenoma detection along with cervical sonography.

The use of ^18^F-choline PET/ceCT for the initial assessment remains debatable, mostly because its cost is much higher than that of MIBI scintigraphy and the tracer availability is much lower than that of Tc99m-MIBI. Furthermore, cervical sonography is a radiation free, inexpensive, and readily available examination. Nonetheless, the high performance of ^18^F-choline PET/ceCT as a second-line approach is promising, and its performance as first-line assessment needs to be investigated in further studies [32]. Our study only included patients with inconclusive findings or negative tumour localisation after MIBI scintigraphy and sonography. Routine first-line use of ^18^F-choline PET/ceCT would probably have higher Se and may simplify the examination of patients with suspected hyperparathyroidism.

The main limitation of our study was the extrapolation of the number of TN glands. Some of the patients benefited from a minimally invasive approach that did not permit the exploration of all four glands. Therefore, we had to extrapolate the number of TN cases by considering glands at unexplored sites as TN glands. However, our study did not show any uncorrelated glands in patients undergoing minimally invasive surgery, thus providing high confidence in this extrapolation method. Hence, Sp and NPV cannot be interpreted as definitive results but rather as an approximation of the ^18^F-choline PET/ceCT performance. There is no randomised controlled trial comparing ^18^F-choline PET/ceCT and cervical sonography or MIBI scintigraphy or different imaging methods, and only one prospective study has investigated this topic [22]. The retrospective design of our study with few selection criteria and prospective data collection shows the real-life application of ^18^F-choline PET/CT. Our study considered only the localisation results, and those regarding biochemical cures might be questionable. We considered that a normalisation of the calcium level at 6 weeks was a very good indicator of cured hyperparathyroidism, mainly because the PTH level can remain high for several months despite hyperparathyroidism being cured.

## 5. Conclusions

In our study, ^18^F-choline PET/ceCT showed excellent diagnostic performance as a second-line imaging modality. The SUVmax, lesion size, and unilocularity seem to be good predictive factors of the correlation between ^18^F-choline PET/ceCT and surgical findings. This imaging method allows localisation of parathyroid adenomas in cases of discrepancy between MIBI scintigraphy and cervical sonography, recurrent or persistent hyperparathyroidism, and ectopic adenomas, especially in the mediastinum. The use of ^18^F-choline PET/ceCT might be extended to first-line assessments, as a replacement for MIBI scintigraphy and/or cervical sonography. Further studies should investigate the performance of ^18^F-choline PET/CT as a first-line imaging approach.

## Figures and Tables

**Figure 1 cancers-14-02029-f001:**
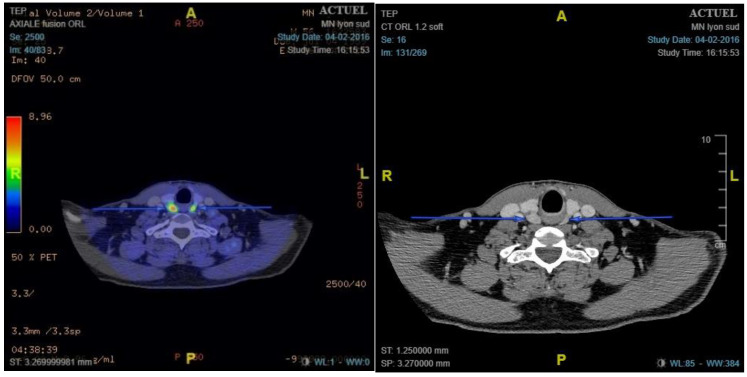
Example of true positive bilateral pathologic gland. ^18^F-choline positron emission tomography/contrast-enhanced computed tomography shows two nodular areas with hypermetabolism: left maximum standardised uptake value (SUVmax), 5.16; right SUVmax, 7.10. Pathology showed two adenomas: right weighing 4620 mg and left weighing 610 mg.

**Figure 2 cancers-14-02029-f002:**
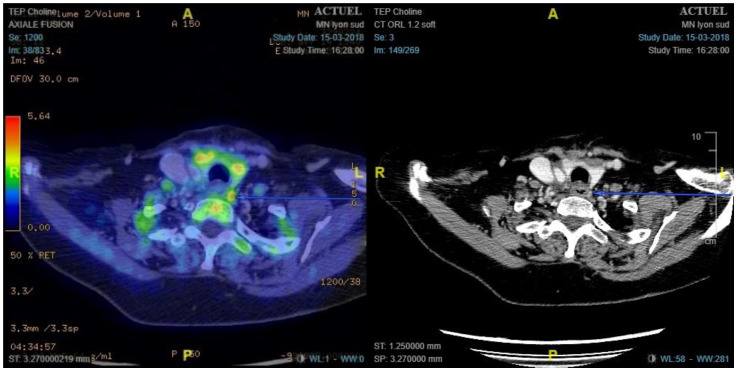
Example of false positive left paraoesophageal lymph node fixing ^18^ F-choline. ^18^F-choline positron emission tomography/contrast-enhanced computed tomography shows left latero-oesophageal nodular hypermetabolism (maximum standardised uptake value, 6.41). Pathology showed a lymph node 1 cm in size.

**Figure 3 cancers-14-02029-f003:**
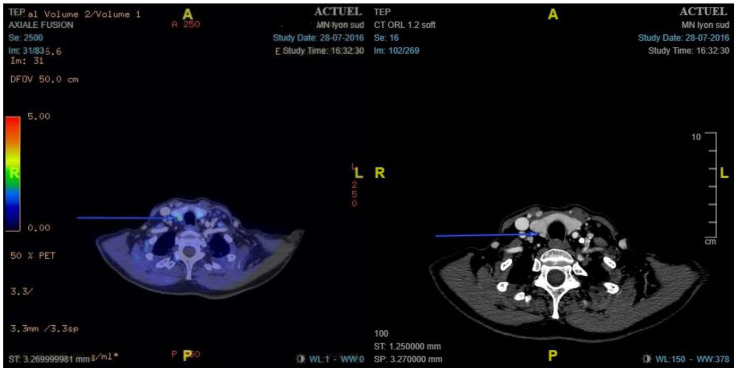
Example of false negative right retro-thyroid gland with insufficient ^18^F-choline fixation. ^18^F-choline positron emission tomography/contrast-enhanced computed tomography shows no nodular hypermetabolism. Pathology showed hyperplasic parathyroid weighing 80 mg.

**Figure 4 cancers-14-02029-f004:**
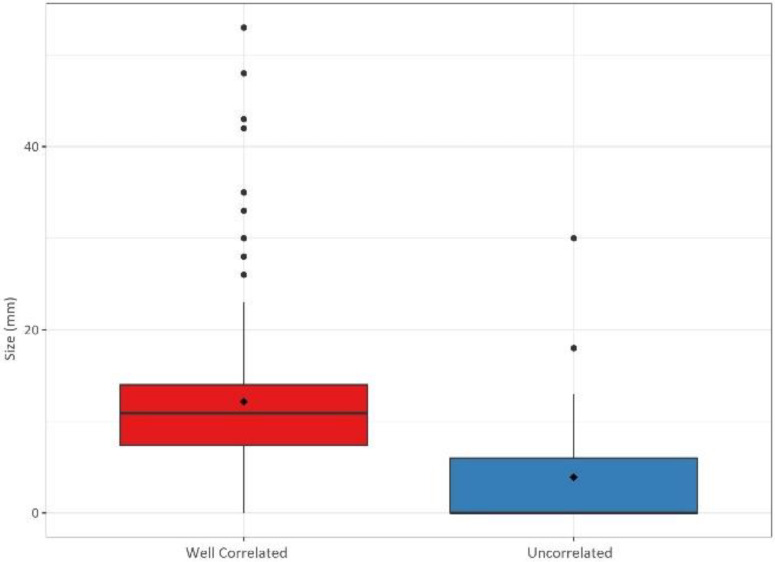
Gland sizes (in mm) in the well-correlated group (*n* = 220) and uncorrelated group (*n* = 49).

**Figure 5 cancers-14-02029-f005:**
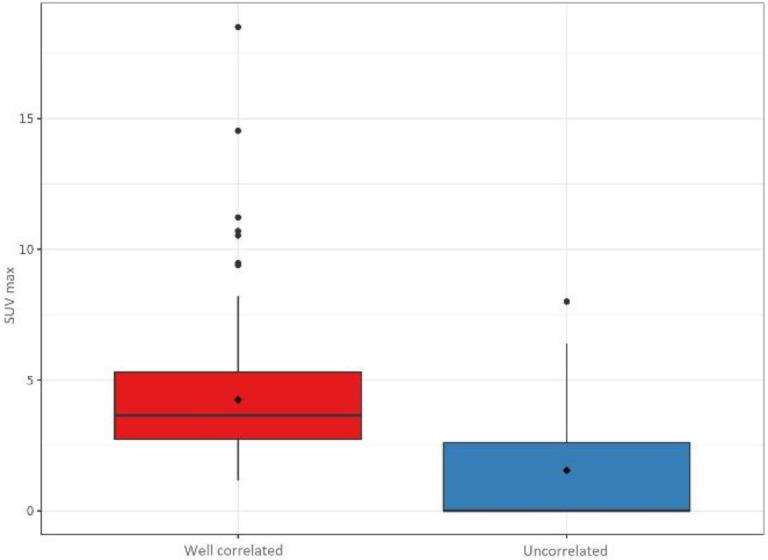
Maximum standardised uptake values (SUVmax) in the well-correlated group (*n* = 220) and uncorrelated group (*n* = 49).

**Figure 6 cancers-14-02029-f006:**
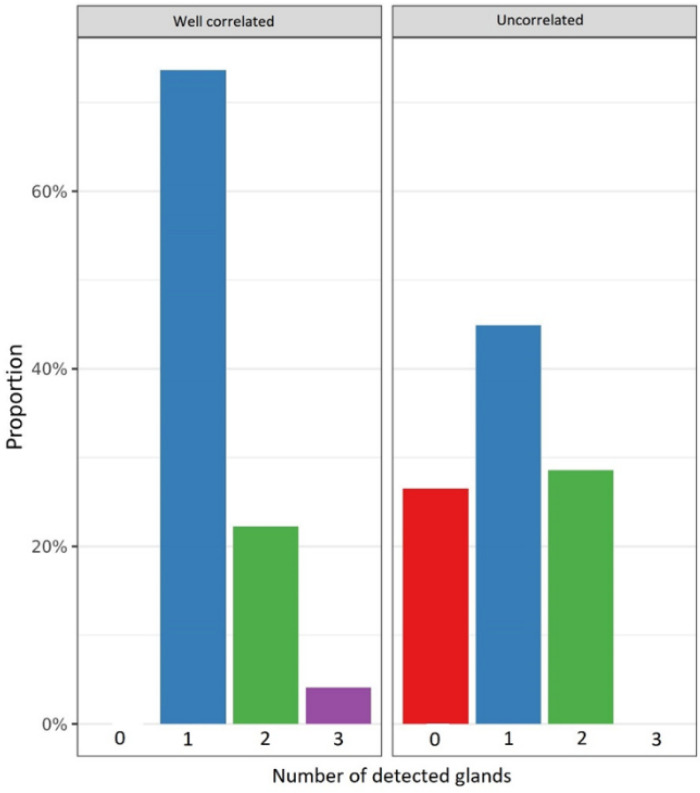
Number of glands detected by ^18^F-choline positron emission tomography/contrast-enhanced computed tomography in the well-correlated group (*n* = 220) and uncorrelated group (*n* = 49).

**Table 1 cancers-14-02029-t001:** Population characteristics.

Characteristics	Mean (Standard Deviation) or Number (Proportion)	Range
Age (years) (*n* = 215)	62 (14.2)	(18–87)
Sex (*n* = 215)		
Male	52 (24%)	
Female	163 (76%)	
BMI (kg/m^2^) (*n* = 215)	26.2 (6.0)	(13.9–51.3)
Hypertension (*n* = 215)	81 (38%)	
Osteoporosis (*n* = 215)	84 (39%)	
Fractures (*n* = 215)	27 (13%)	
Nephrolithiasis (*n* = 215)	63 (29%)	
Previous cervical surgery (*n* = 215)		
Thyroidectomy		
Partial	7 (3.3%)	
Total	7 (3.3%)	
Parathyroidectomy		
Minimally invasive	7 (3.3%)	
SBPE	23 (11%)	
Preoperative markers		
Calcium (mmol/L) (*n* = 215)	2.77 (0.19)	(2.28–3.64)
PTH (normal ratio) (*n* = 208)	2.18 (1.00)	(0.61–19.80)
Postoperative markers, day 1		
Calcium (mmol/L) (*n* = 214)	2.38 (0.196)	(1.930–3.120)
PTH (ng/L) (*n* = 201)	13.9 (10.8)	(4.0–65.0)
PTH (normal ratio) (*n* = 200)	0.35 (0.26)	(0.10–1.94)
Postoperative markers, week 6		
Calcium (mmol/L) (*n* = 209)	2.370 (0.141)	(1.190–2.910)
PTH (normal ratio) (*n* = 195)	0.917 (0.472)	(0.280–4.630)
Surgical approach (*n* = 215)		
Minimally invasive	52 (24%)	
SBPE	163 (76%)	
Surgical complications (*n* = 215)		
Hypocalcaemia	7 (3.4%)	
Recurrent laryngeal nerve palsy	6 (2.9%)	
Superficial hematoma	1 (0.4%)	
Cure rate at 6 weeks (*n* = 210)	202 (95%)	

BMI, body mass index; PTH, parathyroid hormone; SBPE, standard bilateral parathyroid exploration.

**Table 2 cancers-14-02029-t002:** Per-lesion and per-patient contingency tables of the ^18^F-choline positron emission tomography/contrast-enhanced computed tomography results.

	TP	TN	FP	FN	Se	Sp	PPV	NPV
Per lesion	219	592	21	28	88.6%	96.6%	91.4%	95.4%
Per patient	164		21	25	86.8%		88.6%	

TP, true positive; TN, true negative; FP, false positive; FN, false positive; Se, sensitivity; Sp, specificity; PPV, positive predictive value; NPV, negative predictive value; Pr, prevalence. $Se=\frac{TP}{TP+FP}$; $Sp=\frac{TN}{TN+FN}$; $PPV=\frac{Se*Pr}{\left(Se*\mathrm{Pr}+\left(1-Sp\right)\left(1-Pr\right)\right)}$; $NPV=\frac{Sp*\left(1-Pr\right)\;}{\left(\left(1-Sp\right)*Pr+Sp\left(1-Pr\right)\right)}$.

**Table 3 cancers-14-02029-t003:** Correlation between ^18^F-choline positron emission tomography/contrast-enhanced computed tomography and surgical findings.

	Well Correlated (*n* = 220)	Uncorrelated (*n* = 49)	*p*	Test
Age (years)	62.0 ± 14.3	61.5 ± 13.6	0.83	Welch
BMI (kg/m²)	26.2 ± 6.2	26.5 ± 6.0	0.77	Welch
Sex				
Male	51 (23%)	12 (24%)	0.84	χ^2^
Female	169 (77%)	37 (76%)		
Hypertension	86 (39%)	17 (35%)	0.57	χ^2^
Diabetes	14 (6.4%)	2 (4.1%)	0.74	Fisher
Osteoporosis	87 (40%)	18 (37%)	0.72	χ^2^
Fractures	28 (13.0%)	3 (6.1%)	0.19	
Nephrolithiasis	63 (29%)	15 (31%)	0.78	χ^2^
Previous cervical surgery				
Thyroidectomy			0.81	Fisher
Partial	7 (3.3%)	2 (4.0%)		
Total	8 (3.6%)	1 (2.0%)		
Parathyroidectomy				
Minimally invasive	4 (1.8%)	3 (6.1%)	0.093	Fisher
SBPE	20 (9.1%)	7 (14.0%)		
Recurring disease	16 (7.3%)	7 (14.0%)	0.16	Fisher
Persisting disease	8 (3.6%)	3 (6.1%)		
Preoperative markers				
Calcium (mmol/L)	2.76 ± 0.21	2.77 ± 0.18	0.67	Welch
PTH (normal ratio)	2.27 ± 2.18	1.95 ± 0.95	0.12	Welch
Surgeon				
JCL	147 (67%)	36 (75%)	0.27	χ^2^
LM	73 (33%)	12 (25%)		
Surgical approach				
Minimally invasive	52 (24%)	0 (0%)	<0.001	χ^2^
SBPE	168 (76%)	49 (100%)		
PET data				
Size (mm)	12.20 ± 7.61	3.90 ± 6.02	<0.001	Welch
SUVmax	4.25 ± 2.25	1.55 ± 2.00	<0.001	Welch
Number of lesions				
0	0 (0%)	13 (27%)	<0.001	Fisher
1	162 (74%)	22 (45%)		
>1	58 (26%)	14 (29%)		
Cure rate	212 (97%)	43 (91%)	0.081	Fisher

Univariate per-lesion analysis; data are presented as means ± standard deviations or numbers (proportions). BMI, body mass index; PET, positron emission tomography; PTH, parathyroid hormone; SBPE, standard bilateral parathyroid exploration; SUVmax, maximum standardised uptake value; χ^2^, chi-square.

**Table 4 cancers-14-02029-t004:** Correlation between ^18^F-choline positron emission tomography/contrast-enhanced computed tomography and surgical findings.

Variables	OR (95% CI)	*p*	Variables	OR (95% CI)	*p*	Variables	OR (95% CI)	*p*
SUVmax of lesion	0.379 (0.281, 0.510)	<0.001	Size of lesion	0.703 (0.631, 0.785)	<0.001	Number of lesions	0.151 (0.039, 0.497)	<0.01
Osteoporosis	0.854 (0.347, 2.10)	0.73	**-**	1.430 (0.578, 3.56)	0.44	-	0.851 (0.312, 2.210)	0.74
Calcium level	1.140 (0.187, 6.96)	0.89	**-**	1.390 (0.158, 12.2)	0.77	-	0.885 (0.665, 1.150)	0.38
PTH ratio	1.060 (0.741, 1.53)	0.74	**-**	0.986 (0.593, 1.64)	0.96	-	0.784 (0.436, 1.150)	0.35
Age	0.989 (0.958, 1.02)	0.47	**-**	0.985 (0.953, 1.02)	0.37	-	0.991 (0.960, 1.020)	0.56
BMI	0.984 (0.928, 1.04)	0.60	**-**	1.050 (0.983, 1.12)	0.15	-	0.999 (0.921, 1.080)	0.99

Three multivariate per-lesion analyses for SUV max and lesion size and per-patient analysis for number of lesions were conducted by adjusting for the confounding factors BMI, age, osteoporosis, calcium level, and PTH ratio. BMI, body mass index; CI, confidence interval; OR, odds ratio; PTH, parathyroid hormone; SUVmax, maximum standardised uptake value.

## Data Availability

The data presented in this study are available on request from the corresponding author.

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
