# Peer review of "Diagnostic Performance of ^18^F-Choline Positron Emission Tomography/Contrast-Enhanced Computed Tomography in Adenoma Detection in Primary Hyperparathyroidism after Inconclusive Imaging: A Retrospective Study of 215 Patients"

_cancers, 2022, doi:10.3390/cancers14082029_

Round 1
Reviewer 1 Report
The work by Benjamin et al. represents the largest retrospective analysis on the diagnostic effectiveness of 18F-Choline PET/CT thus far. Their results confirm the excellent sensitivity of this method as second-line imaging.
The manuscript is generally clear and well-written. The subject is relevant and current; the implications of these data have a potential to have a significant clinical impact.
There are, however, some concerns, which should be addressed by the authors before publication of this manuscript can be recommended.
Title
Since your imaging protocol included contrast medium, I think that the title should reflect this information (e.g., use PET/ceCT instead of PET/CT).
Simple summary
Line 21. The fact that 18F-Choline PET/CT might replace cervical ultrasound is highly speculative, since neck ultrasonography is an unexpensive, radiation-free and accurate first-line examination. It is, moreover, easily accessible for the patients and can provide information on both the morphology and the vascularization of any thyroid/parathyroid/lymph node finding.
However, Choline-PET could effectively replace its single-photon counterpart, since it features higher resolution and lower acquisition time.
I suggest editing this sentence accordingly.
Abstract
Line 32: Please check the unit of measure
Line 36-39. The sentence is very convoluted and does not clarify which surgical outcome correlates with SUVmax and size (hyperfunctioning adenoma?). Please consider, if it represents your findings, the following edit: “On the multivariate analysis, SUVmax and size correlated with the finding of hyperfunctioning adenoma on pathology”
Line 37: These results depict the univariate analysis. What about the multivariate one?
Methods
Were patients unfit for contrast medium (renal failure, allergy…) excluded from the analysis? Or were they subjected to a protocol without contrast? Please modify the inclusion/exclusion criteria and the description of the methods accordingly.
Results
Multivariate analysis: I find Table 4 unclear: why do the parameters in the rows 3-7 have multiple OR/p-values? Were three different models attempted (one including SUV, one with lesion size, and one with the number of lesions)? In that case, you should change the wording from analysis to analyses and explain why didn’t you build a model with all of the variables (were some of them collinear?). There is also a typo in the table description at the bottom.
Discussion/Conclusions
Again, I disagree on the point that PET might, in spite of its excellent diagnostic features, replace neck ultrasound (please see my comment above). However, it might represent a good candidate to replace 99mTc-Sestamibi SPECT/CT. I suggest you modify the discussion on this matter.
Figures
It would be interesting to add some images depicting the imaging findings in the various categories of clinical situation encountered in the follow up (e.g., images of true positive, false positive, etc.).
Reviewer 2 Report
Well written, well-presented manuscript.
The study is adequately described and clear data presented. Although the study could be further improved by comparing the diagnostic performance with e.g. cervical sonography. Please also address the cost-effectiveness of the technique compared to dual energy CT,
Reviewer 3 Report
The authors retrospectively evaluated the diagnostic performance of 18F-choline PET/CT in the identification of hyperfunctioning adenomas, in primary hyperparathyroidism, after inconclusive imaging, in a series of 215 patients.
On the basis of the results obtained, 18F-choline PET/CT has confirmed to be highly performing as a second-line procedure.
The article is well structured in its entirety.
However, there are considerations to be made.
There is a vast literature that has documented extremely high sensitivity and diagnostic accuracy values for MIBI scintigraphy in the identification of hyperfunctioning adenomas in primitive hyperparathyroidism, especially if conventional planar examination is integrated with SPECT/TC.
In this regard, I suggest you consider the following review in the revised form of the manuscript (SPECT/CT in hyperparathyroidism; Clinical and Translational Imaging, volume 2, Issue 6, pages 537-555, 10 December 2014).
Thus, MIBI scintigraphy, eventually integrated with SPECT/CT, remains the standard radioisotopic imaging procedure of reference. This should be clearly emphasized both in the introduction and in the discussion section.
In the manuscript, the limits of 18F-choline PET/CT should also be better highlighted, in particular as regards availability (much lower than MIBI scintigraphy) and costs (much higher than MIBI scintigraphy).On the basis of the above considerations, I do not consider it appropriate to suggest that the use of 18F-choline PET/CT may be extended “to replace cervical sonography and/or Tc-99-MIBI SPECT/CT as first-line imaging methods in patients with primary hyperparathyroidism” (see lines 21-22 and lines 359-362). These sentences should be deleted.
Round 2
Reviewer 1 Report
Thank you for addressing all my comments. I have no further concerns on your study.
Reviewer 3 Report
No further changes are required